

# Drivers and impacts of westerly moisture transport events in East Africa

Robert Peal[1] and Emily Collier[1]

[1]Department of Atmospheric and Cryospheric Sciences (ACINN), University of Innsbruck, Innsbruck, Austria

**Correspondence:** Robert Peal (Robert.peal@uibk.ac.at)

**Abstract.** Equatorial East Africa (EEA) experiences strong intraseasonal precipitation variations; developing understanding of the processes that drive these variations can improve predictability and help local populations be better prepared for extremes. Previous research has highlighted anomalous westerly moisture transport from the Congo basin as an important driver of enhanced precipitation in EEA. Here, we have developed the first spatially unconstrained, objective framework to detect what we refer to as westerly moisture transport events (WMTEs) in ERA5 reanalysis data from 1980 to 2022, revealing new insights into the drivers of these westerlies and their impact on EEA precipitation. We show that over EEA, WMTEs were most common in January and February between about $5°$ S and $15°$ S, where there were typically 4–6 WMTE days per month, with each individual event persisting for around 2–4 days. During the March to May wet season in EEA, there were on average around 1–2 WMTE days per month. Using a precipitation attribution algorithm, we estimate that WMTEs were associated with up to 60 % of precipitation during January and February in Tanzania, and up to 20 % of precipitation during March–May to the East of Lake Victoria. Consistent with previous work, we found that WMTEs were more likely during phases 2–4 of the Madden-Julian oscillation (MJO). We expand on previous case-study based investigations by showing that the presence of a tropical cyclone anywhere in the south-west Indian Ocean made WMTEs up to three times more likely, even during inactive or unfavourable phases of the MJO. This work builds on previous studies of the westerly wind feature by providing an objective framework to describe EEA westerlies and joins previous work in highlighting the complex nature of the interactions between different features of tropical meteorology that drive these short timescale variations.

## 1 Introduction

Precipitation in Equatorial East Africa (EEA) displays strong intra-seasonal variability, with even the core rainy seasons characterised by alternating periods of wetter and drier conditions (Camberlin and Wairoto, 1997; Pohl and Camberlin, 2006). At their extremes, these wet and dry spells lead to drought and flooding events (e.g., Wainwright et al., 2021)) that can devastate the lives and livelihoods of local populations (e.g., Jones, 2020; Parry et al., 2013; Wright, 2024).

EEA experiences strong hygric seasonality, with the main and secondary wet seasons from March to May (MAM) and October to December (OND), referred to as the long and short rains, respectively; and, the main and secondary dry seasons from June to September (JJAS) and January to February (JF), respectively. Intraseasonal precipitation variability has been mainly studied for the MAM season and has been linked to the activity of the Madden Julian Oscillation (MJO), which is





characterised by an eastward propagating disturbance in zonal winds, surface pressure, convection, and precipitation with an average periodicity of 30 to 60 days (Madden and Julian, 1994). Previous research has shown that precipitation is enhanced over the East African highlands as the centre of convection traverses the Indian Ocean (phases 2–4) and suppressed as it traverses the Pacific Ocean (phases 6 to 8; e.g., Pohl and Camberlin, 2006; Hogan et al., 2015; Finney et al., 2020). The key
mechanism through which the MJO leads to enhanced and even extreme rainfall over the Highlands is the development of anomalous westerly winds that advect moisture from the Congo basin into the region and lead to moisture convergence and deep convection (e.g., Nakamura, 1968; Camberlin and Wairoto, 1997; Pohl and Camberlin, 2006; Finney et al., 2020; Walker et al., 2020).

Until recently, the MJO was the only form of intraseasonal atmospheric variability associated with these anomalous west-
erlies. However, recent case studies of extreme seasons have also linked them to tropical cyclone (TC) activity in the western Indian Ocean (Kilavi et al., 2018; Collier et al., 2019; Finney et al., 2020; Walker et al., 2020; Kebacho, 2024; Gudoshava et al., 2024). Finally, in the most comprehensive, and only multi-decadal, study to date, Finney et al. (2020) found that the westerly circulation during MAM was more common not only in MJO phases 2–4 but also when a TC was located to the north-east of Madagascar, but found no evidence that TCs can act independent triggers. However, this study focused only on
MAM and relied on area-averaged zonal wind anomalies over a fixed region centred over Lake Victoria to identify events. As a result, several aspects of the westerlies remain poorly understood, including their exact spatial location and extent, seasonal characteristics outside of MAM, and the long-term nature of their relationship with the MJO and TCs in driving precipitation variability in EEA.

To address these methodological and knowledge gaps, in this study we present a spatially flexible and systematic framework
for detecting what we refer to as 'westerly moisture transport events' (WMTEs) in the field of moisture transport over eastern Africa. Systematic frameworks for more objective studies of moisture transport exist for atmospheric rivers (ARs). However, existing algorithms largely exclude purely zonal AR-like structures by requiring components of poleward moisture transport (Guan and Waliser, 2015) and, despite recent improvements, detect ARs over eastern Africa on less than 1 % of days (Guan and Waliser, 2024). Using the generated time-series of WMTEs as well as a precipitation attribution algorithm, we present the
first analysis of these events considering all seasons, revealing new insights into their characteristics and their multi-decadal impacts on precipitation in EEA. These insights will contribute to better forecasting of intraseasonal variations, allowing local populations to be better prepared for extreme events.

The algorithm for detecting WMTEs is described in Sect. 2.2. Then, in Sect. 3.1, we present the first analysis of the basic characteristics of these events including the seasonal cycle and typical duration of WMTEs. In Sect. 3.2, we show how the
frequency of WMTEs is altered by MJO phase and the presence of TCs, and then finally in Sect. 3.3, we use the time-series of WMTEs to estimate how much EEA precipitation can be attributed to westerlies in each season.




## 2 Data and methods

### 2.1 Data

For our detection algorithm and analysis of regional circulation, we used the ERA5 reanalysis, which is provided by the
European Center for Medium-Range Weather Forecasts on a $0.25° \times 0.25°$ latitude and longitude grid from 1940 to present
(Hersbach et al., 2020). For detection, we used daily moisture transport fields at 700 hPa, which we calculated using the
product of the daily averaged wind vector $\mathbf{u}$ and the daily averaged specific humidity $q$. We focused on the 700 hPa level as it
is correlated with both sub-seasonal and interannual variations in precipitation during the long rains (Walker et al., 2020).

We quantified MJO activity using the Australian Bureau of Meteorology's all-season real-time multivariate MJO index
(Wheeler and Hendon, 2004), which provides MJO phase and amplitude information from 1979 to the present day (Bureau
of Meteorology (BoM), Accessed 06.02.2024). We found that the patterns of WMTE activity were similar in phases 2–4,
and 5–1, which previous research identified as favourable and unfavourable, respectively, for westerly winds and enhanced
precipitation in EEA (Pohl and Camberlin, 2006; Finney et al., 2020). Therefore, when investigating the role of the MJO in
WMTE characteristics, we grouped our analysis into favourable, unfavourable, and inactive phases of the MJO.

We quantified TC activity using the International Best Track Archive for Climate Stewardship global storm archive (IB-
TrACS) Knapp et al. (2010). IBTrACS provides information about TC position, intensity, and structure at three-hourly tem-
poral resolution since 1842. We used the archived storm data from both the northern and southern Indian Ocean basins. We
used IBTrACS reports from mid-day to aggregate the data to daily timescales. IBTrACS may suffer from missing events prior
to the availability of satellite observations in the 1960s, and from temporal heterogeneities in the methods used for estimating
intensity prior to the 1980s (Kossin et al., 2013). Therefore, we focused our analysis on the period from 1980 to 2022.

We investigated the precipitation response to WMTEs using daily precipitation totals provided by ERA5. Due to potential in-
accuracies in ERA5 precipitation totals (Lavers et al., 2022), we compared the ERA5 precipitation response with that from two
other precipitation products: the Climate Hazards Group InfraRed Precipitation with Station data (CHIRPS), which provides
daily precipitation totals at $0.05° \times 0.05°$ resolution, obtained through interpolation of satellite and rain gauge observations
from 1998 to the present (Funk et al., 2015); and, Integrated Multi-satellitE Retrievals for GPM (IMERG), which provides
daily precipitation totals at $0.1° \times 0.1°$ from infra-red and microwave satellite observations, from 1998 to the present (Huffman
et al., 2023).

The area used in our detection algorithm included all of Africa and the Indian Ocean, extending from $22°$ W to $102°$ E and
from $40°$ N to $37°$ S. However, we focus our results and discussion on the regions of eastern Africa and south-west Indian
Ocean where our phenomenon of interest occurs.

The detected WMTE masks, and the code for detecting WMTEs, are available in a dataset at https://doi.org/10.5281/zenodo.
15173985 (Peal and Collier, 2025).



## 2.2 Detection of westerly moisture transport events

We identified westerly events using a method adapted from a detection algorithm for atmospheric rivers (Guan and Waliser,
2015). Following this approach, we first identified grid points in the field of daily mean moisture transport where **(i)** the
direction of the moisture transport vector was within $45°$ of westerly, and **(ii)** the magnitude of moisture transport exceeded
the 70th percentile of magnitudes recorded at that location, for that month, from 1980 to 2022. We then identified contiguous
regions comprising at least 1000 ERA5 grid points, approximately 560,000 km$^2$ at the equator, satisfying these conditions. In
this manuscript, we refer to the identified events as 'westerly moisture transport events' (WMTEs).

To determine whether a given WMTE was associated with one or more TCs in the Indian Ocean, we used a 500-km dis-
tance threshold between a storm's position and any part of the polygon outlining the extent of the WMTE. When discussing
TC attribution in this manuscript, the subset of events associated with TCs are referred to as TC-WMTEs, while all other
events are referred to as noTC-WMTEs. While the mechanism by which TCs and westerlies in EEA may be related remains
unconstrained, we included this distance threshold because in several of the case studies that have shown TCs and westerlies
coinciding (e.g., Collier et al., 2019; Kilavi et al., 2018), the westerly feature appears to be joined to the northern part of the
vortex (see also Fig. 1a). The distance threshold therefore represented a simple way to identify events with this configuration.

Figure 1 shows an example of a detection of a TC-WMTE, on 9 January 2015. Westerly moisture transport is visible across
EEA at around $10°$ S in the 700 hPa moisture flux field, while a TC is located off the coast of Madagascar (Fig. 1a).

We defined events affecting EEA as events where the WMTE polygon intersected the line extending from $2°$ N to $12°$ S at
$29°$ E, shown by the dashed line in Fig. 1a. There is little consensus on the exact extent of EEA (Zaitchik, 2017). We chose
$29°$ E such that Burundi, Kenya, Tanzania, and Uganda – the westernmost countries considered part of EEA – lie to the east
of the line, while most of the Congo lies to the west. With this delineation, WMTEs intersecting this line by definition import
moisture from the Congo basin into EEA. We selected the latitude range to capture both the region where WMTE occurrence
is highest (see Sect. 3.1) and the region used in Finney et al. (2020) to define westerly events.

As we found reasonably consistent patterns of WMTE occurrence within the different hygric seasons in EEA, we present
our results grouped by JF, MAM, JJAS and OND.

## 2.3 Attribution of precipitation

Precipitation was attributed to WMTEs following the methods of Konstali et al. (2024). For this approach, precipitation was
first masked to only include grid cells with at least 1 mm per day, a threshold commonly used to define a 'wet day' (Klein Tank
et al., 2009). Precipitation was then divided into contiguous areas surrounding local maxima using a watershed approach
(Beucher and Lantuéjoul, 1979; Konstali et al., 2024), with maxima separated by less than 750 km placed in the same polygon.
We detected the precipitation polygons using the implementation in the Dynlib software from Spensberger (2021). Finally,
precipitation polygons (masks) that overlapped with WMTEs were attributed to them. For ERA5, precipitation was attributed
to a WMTE if any grid cell in the precipitation mask was also inside a WMTE mask. To account for the different resolutions of





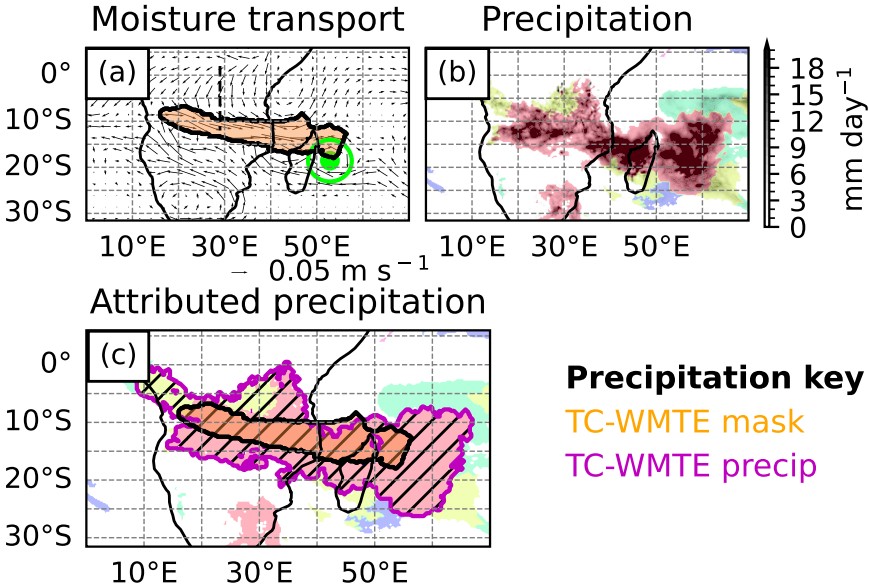

**Figure 1.** An example of WMTE detection and precipitation attribution, on 9 January 2015. **(a)** Daily moisture flux at 700 hPa and the detected WMTE, which coincided with TC Bansi. The WMTE mask is shown in orange. The position of the TC is shown by the green dot while the circle around it shows the 500 km radius. The black dashed line from $2°$ N to $12°$ S at $29°$ E was used to identify WMTEs advecting moisture into EEA. **(b)** ERA5 daily precipitation total. Each differently coloured region represents a different precipitation object. **(c)** The precipitation attributed to the TC-WMTE, identified by overlaying the masks in **(a)** and **(b)**, which is delineated by the purple contours.

CHIRPS and IMERG, precipitation masks were attributed to WMTEs if the nearest ERA5 grid point to any native grid point in the precipitation mask was part of a WMTE mask.

Figure 1b shows the precipitation field and precipitation polygons after the watershed algorithm was applied. All the precipitation in any polygon that overlapped with the WMTE polygon was attributed to the WMTE (Fig. 1c). In this example, a large area of precipitation reaching across Africa into the south-west Indian ocean was attributed to the TC-WMTE.

## 3 Results and Discussion

### 3.1 Characteristics of WMTEs

We first present some basic characteristics of WMTEs in eastern Africa. Over this region, WMTEs were most common in JF: most locations east of $20°$ E between about $5°$ S and $15°$ S experienced a total of around 4–6 WMTE days per month (Fig. 2a) and associated potential moisture advection from the African interior. Most locations in this region experienced around 2

separate events per month (Fig. 2e), with each event persisting for 2–4 days and the longest durations recorded at the coast (Fig.



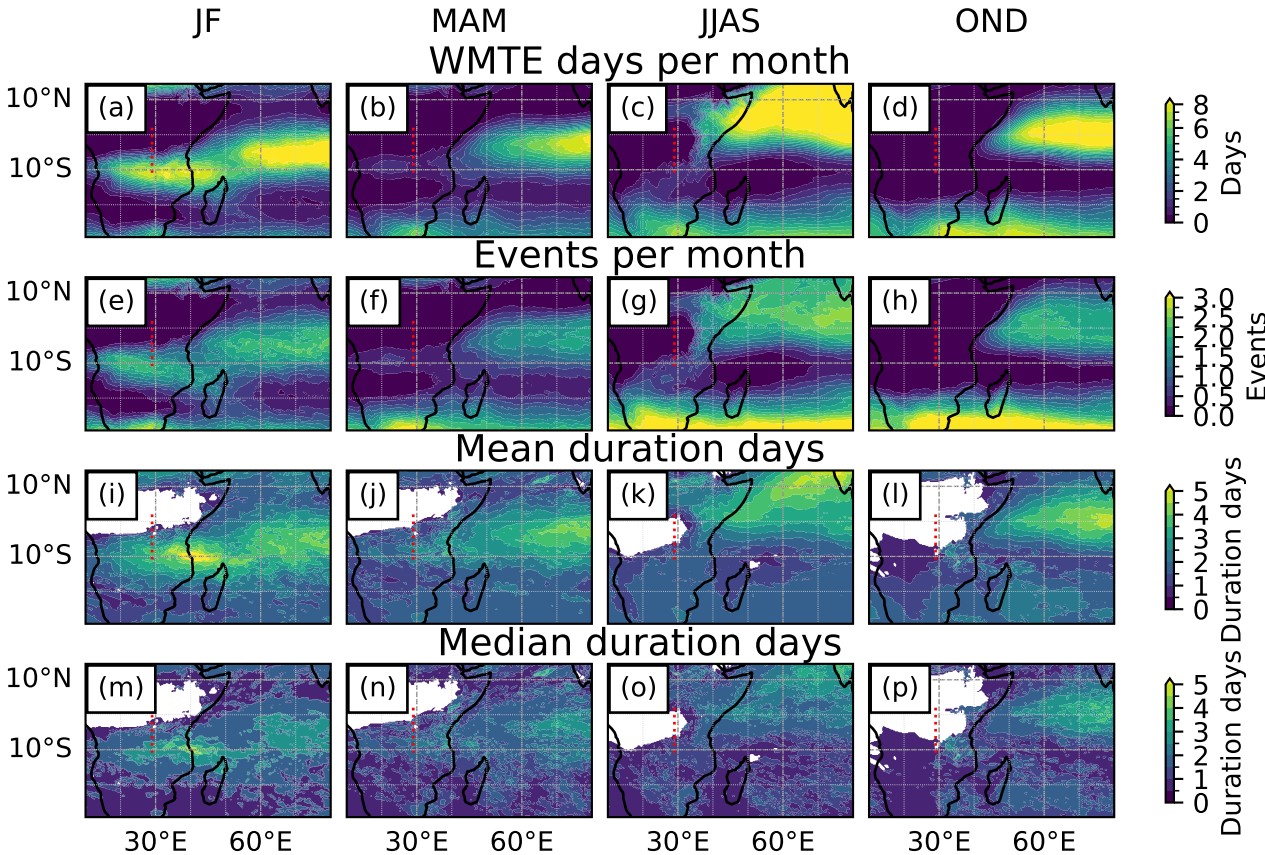

**Figure 2.** Some statistics of WMTEs: **(a–d)** typical number of WMTE days per month. **(e–h)** typical number of WMTEs per month, defined as periods of consecutive days where there was a WMTE present at that location. **(i–l)** mean duration of WMTEs. **(m–p)** median duration of WMTEs, in different seasons. White shading indicates areas where were no events detected and so no mean or median could be calculated.

2i). The median duration (Fig. 2m) in eastern Africa was typically around 1.5 days less than the mean duration, highlighting the prevalence of short events.

In the other months, detected WMTEs in EEA were much rarer. In MAM, there were on average 1–2 WMTE days per month (Fig. 2b). Events only occurred roughly every two months (Fig. 2f) and the mean and median event lengths were just 1–2 days (Fig. 2j, n). However, the area covered by these events means that they would have the potential to advect moisture into EEA from the African interior. In JJAS, there were around 2 events per year at most grid points in the horn of Africa (Fig. 2g), lasting 1–2 days (Fig. 2k, o). However, these events tended not to extend far enough westward to advect moisture into EEA. In OND, there were almost no WMTEs in EEA (Fig. 2d).

For the remainder of the paper, we focus our analysis on WMTEs that affect EEA (i.e. those overlapping with the line at 29° E from 2° N to 12° S, as discussed in Sect. 2.2; Fig. 3). Focusing on JF and MAM, the seasons with at least one





WMTE with its westerly extent near the Congo Basin (Fig. 2a–d), we compare regional moisture transport patterns on days with and without WMTEs affecting EEA. On days without a WMTE affecting EEA, in both JF (Fig. 3a) and MAM (Fig. 3d), moisture transport across Eastern Africa is approximately easterly/north-easterly, as expected. There was a region of anti-cyclonic rotation (positive curl in the southern hemisphere) over southern Africa north of about $20°$ S, and in JF, there was also

a region of cyclonic rotation in the region south of $10°$ S. In the Indian Ocean, in both seasons there were weak westerlies east of Madagascar between the equator and $10°$ S.

On days with a noTC-WMTE crossing the EEA line, the circulation patterns were similar in both JF (Fig. 3b) and MAM (Fig. 3e): north-easterly winds over Africa underwent sharp curvature at about $5°$ S, leading to strong westerlies crossing back through eastern Africa at about $10°$ S, and connecting to the westerlies in the Indian Ocean. Both seasons, similar to Fig. 3a,

show cyclonic rotation south of $10°$ S and anti-cyclonic rotation south of $20°$ S. Particularly in JF, there was also weak cyclonic rotation in the Mozambique channel.

The moisture transport patterns on days with a TC-WMTE crossing the EEA line are similar to the noTC-WMTE case for both JF (Fig. 3c) and MAM (Fig. 3f), but with enhanced cyclonic rotation in the Mozambique channel.

## 3.2    MJO and TCs as drivers of WMTEs

To investigate the role of the MJO and TCs as drivers of WMTEs, we analysed the frequency of WMTEs in different MJO phases and on days with TCs present at different locations in the Indian Ocean. WMTE days occurred at the highest rate over eastern Africa during favourable MJO phases in JF: the region between $6°$ S and $10°$ S from the Congo basin to the Indian Ocean had a WMTE present on around one third of days in these phases (Fig. 4a). In the unfavourable phases and inactive periods, the same region had around 1–2 WMTE days per month (4e, i), suggesting that while WMTEs are more probable in

the favourable phases, MJO phase cannot explain all the variation in WMTE frequency.

Similarly in MAM, WMTEs affected EEA on around 1 of the $\sim7$ favourable phase days per month (Fig. 4b), while events during the other phases were less common (Fig. 4f, j). Furthermore, WMTEs during the favourable phases were more likely to extend further westward where they could advect moisture from the African interior into EEA.

In JJAS, the number of WMTE days per month over EEA was rather similar for all MJO phases, but WMTE days occurred at

a higher rate during favourable phases because there tended to be fewer days in the favourable phases across the whole month (Fig. 4c, g, k). Similarly for OND, locations across Africa between about $7°$ S and $12°$ S experienced roughly one WMTE day every five years during both the favourable and inactive phases, but since inactive phases were almost twice as common as favourable ones, the MJO still increased the rate of WMTE occurrence (Fig. 4d, h, l).

We also investigated whether WMTEs that affected EEA were more frequent on days with TCs present in the Indian

Ocean, and whether certain TC locations enhanced or suppressed the WMTEs. For this analysis, we divided our study region into $5° \times 5°$ grid boxes and in each box calculated the fraction of the days where a TC was present while a WMTE crossed the EEA line. Since it is known that TC activity is itself influenced by the MJO (Balaguru et al., 2021; Liebmann et al., 1994), we further subdivided our analysis by its simplified phase. The presented probabilities can then be described as: $P(\text{WMTE}|\text{TC in box \& MJO phase})$.





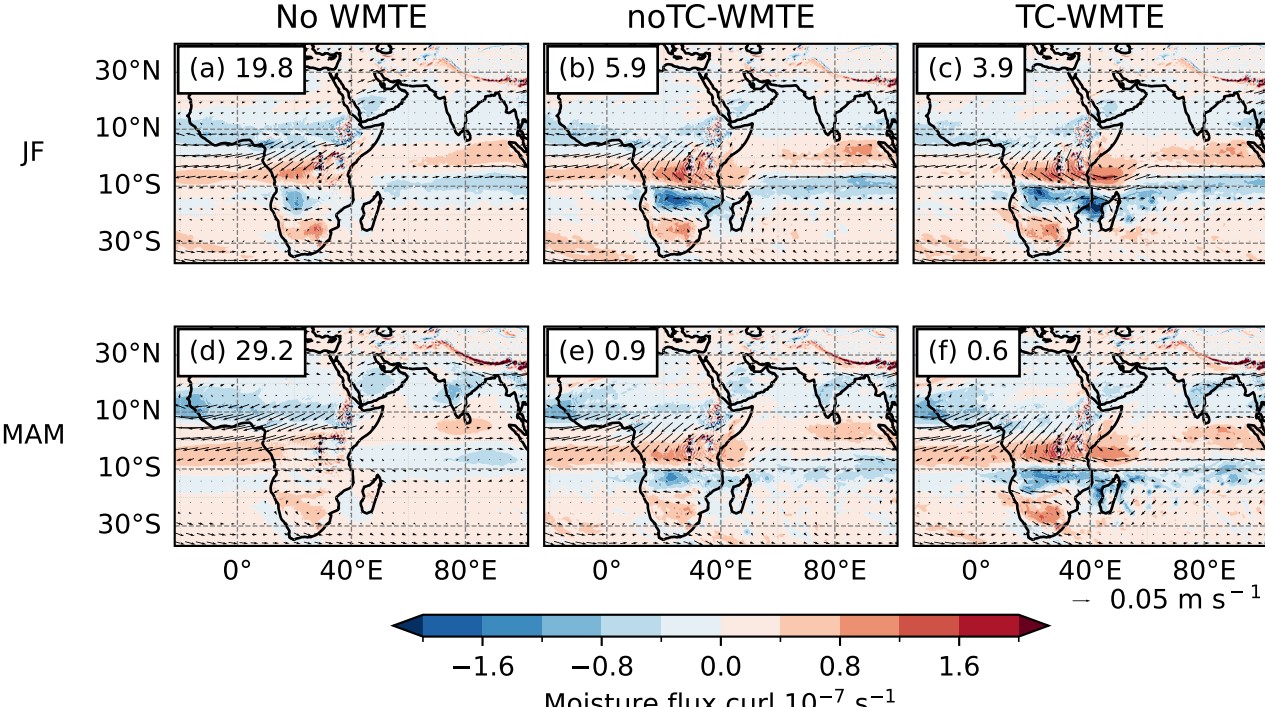

**Figure 3.** Composite of 700 hPa moisture transport in **(a–c)** JF and **(d–f)** MAM. **(a, d)** Days without a WMTE crossing 2° N–12° S along 29° E, shown as the black dotted line. **(b, e)** Days with a noTC-WMTE crossing the EEA line. **(c, f)** Days with a TC-WMTE crossing the EEA line. The shading represents the curl of the moisture transport field. The numbers labelled in each figure show the average number of days per month in that composite

Figure 5 shows the results of the analysis. Since TCs are relatively rare, there were typically only a few TC reports in each box, with the locations with the highest number of reports having around 15–25 TCs in each simplified MJO phase over the study period. While it is difficult to make robust conclusions from this analysis, we describe the general patterns that emerge.

The probability of a WMTE crossing the EEA line given there was a TC present was highest for TCs in the Mozambique channel during favourable MJO phases (Fig. 4a), with WMTEs occurring on roughly 80 % of days where there was a TC at that location. The probability decreased moving east in the southern Indian Ocean, but remained high at around 40 % as far as 65° E. The probabilities in the northern Indian Ocean, as well as for unfavourable phases (Fig. 4b) and inactive MJO conditions (Fig. 4c) were smaller, with a WMTE affecting EEA on ∼10–20 % of days where there was a TC present south of 15° S and west of 60° E. The locations with the highest probabilities were again the Mozambique channel with a rate of around 40 % for inactive MJO and 30 % for unfavourable phases.

In the favourable MJO phases (Fig. 5a), there was a region inland (32.5° E, 17.5–22.5° S) where $(P(\text{WMTE}|\text{box \& MJO phase})$ was almost 1, since when TCs did persist this far inland, the northern section of the vortex crossed the EEA line and was al-





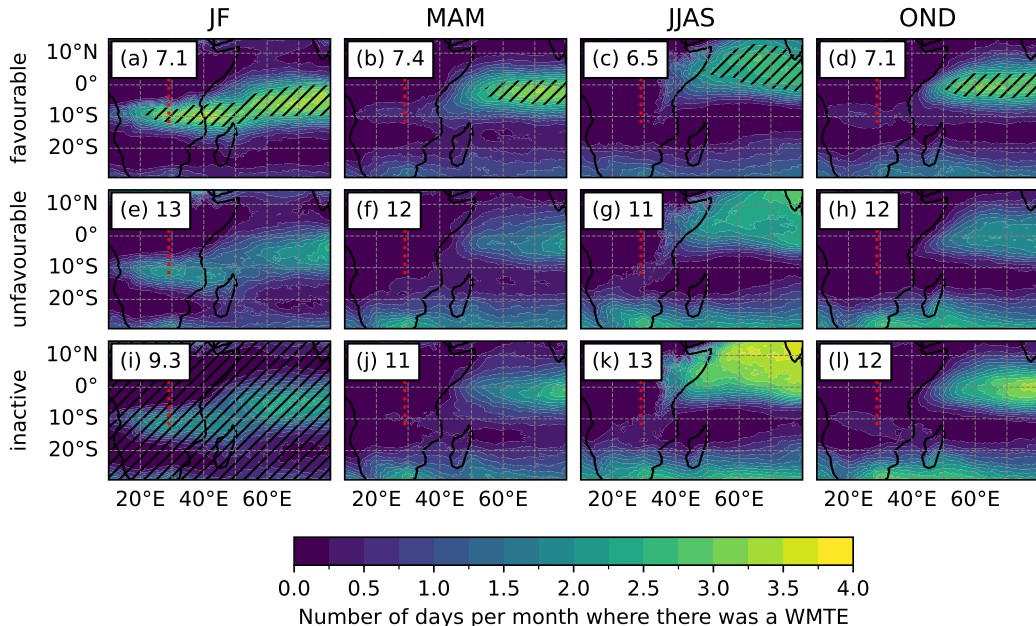

**Figure 4.** The relationship between MJO phase and the number of WMTE days. The number of days per month with a WMTE in each grid box during: **(a, e, i)** JF; **(b, f, j)** MAM; **(c, g, k)** JJAS; **(d, h, l)** OND. **(a–d)** show MJO phases 2–4; **(e–h)** show MJO phases 1 and 5–8, and **(i–l)** show inactive MJO. The number on each panel indicates the average number of days per month in the MJO phases shown in that panel. The hatched areas are where more than one third of the days in the box are WMTE days. The red dashed line shows the 'EEA line'.

ways part of a TC-WMTE mask. However, TCs persisted this far inland less than five times during the whole study period of 1980-2022.

To further disentangle TC impacts from MJO impacts during the peak WMTE season of JF, we also show the risk ratio, also known as relative risk (Simon, 2001), comparing the probability of a WMTE affecting EEA for a given MJO phase for two cases: when a TC was present in a given box and when no TCs were present anywhere in the Indian Ocean (Fig. 5d–f). A risk ratio greater (less) than one indicates that WMTEs affecting EEA occurred at a higher (lower) rate when there was a TC present in that box compared with when there was no TC present in the Indian Ocean, while a ratio of one indicates no effect. Considering the whole year, the risk ratio is extremely high for nearly all locations in the southern Indian Ocean. This is because both WMTEs and TCs in the southern Indian Ocean (Mavume et al., 2010) peak in JF, and so the rate of WMTE occurrence when there was no TC present was extremely low (see Fig. S3). Therefore, to explore the link between TCs and WMTE in more detail, we display this analysis only for JF. The displayed statistic is then:

$$\frac{P(\text{WMTE}|\text{TC in box \& MJO phase \& season=JF})}{P(\text{WMTE}|\text{no TC \& MJO phase \& season=JF})}.$$

For JF, the risk ratio was above 1 in the Mozambique channel in all phases: TCs in this location were associated WMTEs occurring at a rate of 1.5 to 2.5 times the rate of WMTEs without a TC in the favourable and inactive phases, and by up





to 3 times for unfavourable phases. The risk ratio was also above 1 at most locations in the south-west Indian Ocean east of Madagascar. The enhancement was strongest in the unfavourable phases where the risk ratio was often over 2 even as far as $80°$ E. The values are noisy with strong local fluctuations of opposing sign, making it hard to draw strong quantitative conclusions, but the analysis suggests that TC presence is associated with an enhanced rate of WMTEs in most of the south-west Indian
Ocean and in all MJO phases, including inactive MJO.

We repeated this analysis using only the previously defined TC-WMTE in the risk ratio numerator (SI Fig. S4). The results around Madagascar were broadly similar, but the area with risk ratio exceeding 1 only extended to around $55°$ E, because it was rare for WMTEs that crossed the EEA line to extend far enough into the Indian Ocean to be within 500 km of the observed TC location (see e.g. Fig. 2a, where the region with high days per month over EEA extends to around $55°$ E).

This analysis shows that throughout the year, and independent of MJO phase, WMTEs are more likely on days with a TC present almost anywhere in the Indian Ocean, even if the TC is far away from the WMTE location. We defined the concept of TC-WMTEs to account for cases such as that in Fig. 1, where the WMTE was directly attached to the northern part of the TC circulation. However, the finding that the likelihood of WMTEs is enhanced even when TCs are at a greater distance, highlights that the mechanism linking TCs and WMTEs is not as simple as just a connection of the circulation.

While the mechanism causing this link remains unclear, the finding is consistent with case studies of anomalously wet MAM 2018 & 2020 (Kilavi et al., 2018; Gudoshava et al., 2024)) and OND 2006 & 2019 (Collier et al., 2019; Wainwright et al., 2021) seasons. It further shows that over the 43-year study period, TCs anywhere in the vicinity of Madagascar increased WMTE occurrence. This finding is in contrast with a case study of the anomalously dry MAM 2019 (Gudoshava et al., 2024) and the long-term analysis of (Finney et al., 2020), which suggested that TCs to the west of Madagascar may be associated with
reduced precipitation in EEA. However, for MAM 2019, it is worth noting that the largest precipitation totals in EEA during this season were recorded while cyclone Idai was active off the northwest coast of Madagascar. Differences in the long-term patterns of WMTE, MJO and TC interactions likely arise due to differences in methodology, as Finney et al. (2020) identified westerly events based on anomalies of area-averaged zonal velocity over a fixed box over Lake Victoria, while we employ a flexible and objective algorithm that detects westerly moisture transport exceeding a certain spatial scale and magnitude.

We therefore also assessed the robustness of our detection algorithm by performing a sensitivity analysis of the three free parameters: the directional coherency ($\pm45°$), the minimum area (exceeding 1000 ERA5 grid points), and the transport magnitude (exceeding the month-dependent 70th percentile) (see detailed discussion in Sect. S1 of the SI). Over EEA, the number of WMTE days per year was most sensitive to the transport magnitude threshold, decreasing by about 26 % in response to a 10 % increase in the threshold, while changes in the direction and size thresholds had smaller and negligible impacts on the
number of WMTE days per year, respectively.

### 3.3   Precipitation attribution to WMTEs

Finally, we present the amounts and fractions of ERA5 precipitation that were attributed to WMTEs from 1980 to 2022. We note that over the overlapping period where data are available in CHIRPS, IMERG, and ERA5, from 1998 to 2022, similar fractions and spatial patterns of precipitation were attributed in all three datasets (see Sect. S4 of the SI).





**Figure 5.** How the presence of a tropical cyclone at different locations changes the probability of a WMTE day in EEA. **(a–c)** The probability of a WMTE crossing the line at 29° E from 2° N to 12° S, shown by the black dashed line, given the presence of a TC in 5° grid boxes. **(d–f)** JF relative risk for each box, showing the ratio of the probability of a WMTE crossing the line given the presence of a TC, compared to the probability of a WMTE crossing the line, given there is no TC anywhere in the Indian Ocean, for JF. Boxes with a dot have at least 5 TC reports in the period 1980–2022. **(a, d)** MJO phases 2–4, **(b, e)** MJO phases 5–1, **(c, f)** MJO inactive.





Figure 6a–d shows the average amount of precipitation per month at each grid point during each season, contextualising the importance of precipitation attributed to WMTEs compared to other drivers within each season.

     The main wet season south of 5° S, including most of Tanzania, is in JF (Fig. 6a), coinciding with the season where WMTEs were particularly prevalent over EEA (cf. Fig. 2a). During this season, up to 60 % of the precipitation was attributed to WMTEs (Fig. 6), suggesting that moisture transport from the African interior is an important source of moisture for this rainy season.

This result is consistent with Kebacho (2024), who showed that easterly flow during the Tanzanian wet season of JF inhibited precipitation.

     During MAM (Fig. 6b) around 20 % of the precipitation to the south and east of Lake Victoria was associated with WMTEs. Although this is a smaller fraction than in JF, most locations in EEA experienced less than one WMTE day per month in this season (cf. Fig. 4b, f, j), highlighting that WMTEs in MAM can be associated with large precipitation amounts. This result

is consistent with Finney et al. (2020), who found that days with westerly moisture transport in EEA during MAM had up to a 200 % increase in precipitation totals, although we found that precipitation attributed to WMTEs was concentrated in the eastern part of their study region, shown as the red box in Fig. 6b.

     There are some regions where very large fractions of the precipitation were attributed during local dry seasons, such as around 15° N in JF (90 %; Fig. 6e) and 35° E–40° E from 0° N–5° N in JJAS (70 %; Fig. 6g), suggesting that precipitation at

these locations outside the main wet seasons is often associated with WMTEs, but the total amounts of precipitation associated with such events are relatively small over the whole time period.

     We used the framework of Konstali et al. (2024) for precipitation attribution. We note that in their study, much of the precipitation in the tropics including eastern Africa, was unattributed. In that work, atmospheric rivers were detected using moisture transport axes (Spensberger et al., 2024), a technique that is well established for detecting ARs, but mostly at high

latitudes, and they did not detect features over EEA matching our WMTEs. It is possible that WMTEs differ from high latitude ARs in such a way that they were not detectable using the moisture transport axis method. One possibility is that the typical circulation geometry around WMTEs: the easterly circulation reversing to westerly over central Africa (Fig. 3b, e), may mean that there is no clear axis of maximum moisture transport, even though there is strong westerly transport over EEA.

## 4   Conclusions

In this study, we developed the first spatially flexible and objective framework for detecting WMTEs and their impacts on EEA. We used the resulting timeseries of WMTES to perform a multi-decadal study of their drivers and regional impacts, providing new insights including their importance outside of the main rainy seasons. We showed that WMTEs affecting EEA exhibit a strong seasonal cycle, peaking during JF, where they occurred on up to six days per month. There were also a few WMTE days per month during the MAM long rains, and even fewer during the OND short rains. We also demonstrated the long horizontal

extent of these features, with WMTEs over EEA often stretching right across the African continent and sometimes out into the Indian Ocean. On days with a WMTE in EEA, the circulation remains easterly or north-easterly between the equator and about 10° N, but turns sharply to come from the west around 5° S north of the equator.





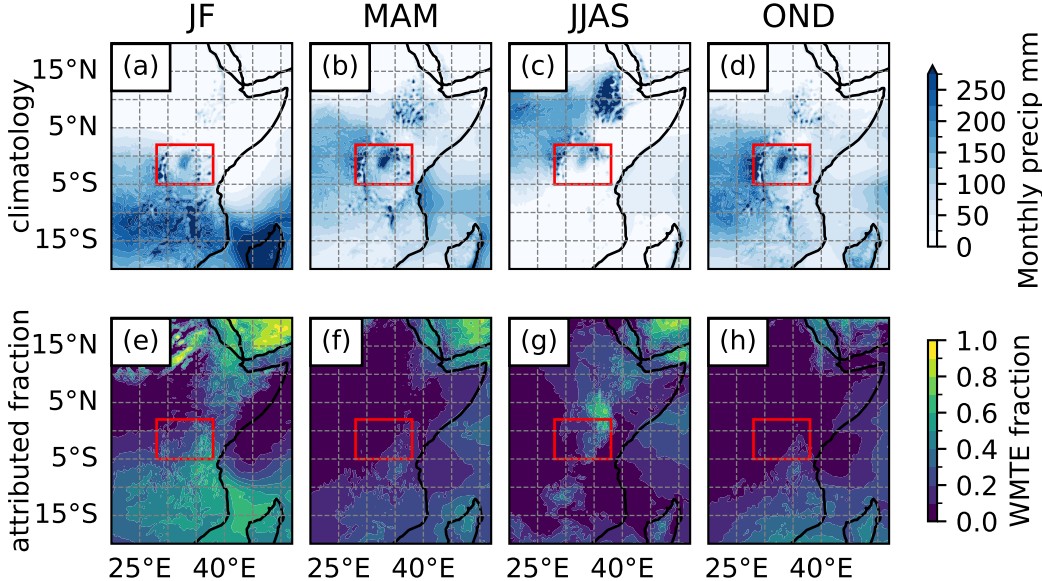

**Figure 6.** (a–d) The ERA5 average monthly precipitation from 1980 to 2022 in each season, and (e–h) the fraction that was attributed to WMTEs in each season. The red box shows the extent of the region studied in Finney et al. (2020).

The seasonal patterns of WMTE frequency were reflected in the amounts of attributed precipitation: up to 60 % of JF precipitation was attributed to WMTEs, highest over Tanzania where these months correspond to the wet season, while up to 270 20 % was attributed in MAM, highest to the south and east of Lake Victoria and over Tanzania.

Consistent with previous research, we showed that WMTEs advecting moisture from the Congo region into EEA are associated with the MJO: during JF, MAM, and OND, WMTEs were more common in EEA during MJO phases 2–4, but still occurred during other MJO phases and when the MJO was inactive. However, our analysis also supports a potential independent role of TCs in triggering or enhancing westerly moisture transport into EEA, which has previously only been identified 275 on a case study basis (e.g., Kilavi et al., 2018; Collier et al., 2019; Gudoshava et al., 2024)).

The timeseries of WMTEs is publicly available, which we hope will facilitate further study of their long-term impacts and drivers as well as case studies of individual seasons or events. Areas for future work highlighted by this research include: investigating the causality chain and exact processes linking WMTEs with TC and MJO activity; assessing which TC properties, such as location and intensity, may affect the association between TCs and WMTEs; and which properties of WMTEs and/or 280 their associated circulation patterns modulate their association with extreme precipitation in EEA.

Our work provides novel insights into WMTE drivers and impacts on EEA and joins previous research in highlighting the importance of ensuring that not only MJO and TCs themselves, but also their interactions, are well represented in weather models for accurate process modelling of sub-seasonal precipitation variability in EEA and for improved forecasting of extremes.



*Code and data availability.* The timeseries of WMTEs and attributed precipitation, as well as code for WMTE detection and precipitation
attribution, and producing the figures in this paper, are available at https://doi.org/10.5281/zenodo.15173985, Peal and Collier (2025). The
Dynlib software used to define the precipitation polygons is available at https://doi.org/10.5281/zenodo.4639624, Spensberger (2021). ERA5
data are freely available from the Copernicus Climate Change Service (C3S) Climate Data Store (https://doi.org/10.24381/cds.adbb2d47,
Hersbach et al. (2023)). IBTrACS tropical cyclone locations are available from https://doi.org/10.1175/2009BAMS2755.1 (Knapp et al.,
2010). BOM MJO indices are available from the LDEO/IRI data library (https://iridl.ldeo.columbia.edu/SOURCES/.BoM/.MJO/.RMM/
phase/index.html).

*Author contributions.* Research design: all authors; Analysis and methodology: RP; funding acquisition: EC; Writing: all authors

*Competing interests.* The authors declare no competing interests.

*Acknowledgements.* This research was funded by the Austrian Science Fund (FWF) [Grant No. P-36624]. Additional support was provided
by the Innsbruck Network for Weather and Climate Research (IWCR). The authors gratefully acknowledge the scientific support and HPC
resources provided by the Erlangen National High Performance Computing Center (NHR@FAU) of the Friedrich-Alexander-Universität
Erlangen-Nürnberg (FAU) under the NHR project c104fa. NHR funding is provided by federal and Bavarian state authorities. NHR@FAU
hardware is partially funded by the German Research Foundation (DFG) – 440719683. For open access purposes, the author has applied a
CC BY public copyright license to any author-accepted manuscript version arising from this submission.



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
