# Peer review of "Drivers and impacts of westerly moisture transport events in East Africa"

_EGUsphere, 2025_

## Author Comment (AC1)

Response to Reviewer 2

Reviewer comments are provided in black font, while our replies are in blue.

This study investigates the prevalence and impact of westerly moisture transport events (WMTEs) on East Africa. The importance of moisture flowing from the Congo has been highlighted in previous studies, but this study applies a novel and more complex approach to identify the WMTEs. The method adapts previous work looking at atmospheric rivers for this tropical context. The method is applied to reanalysis and rainfall observation products. The impacts and wider features considered are rainfall, Madden Julian Oscillation (MJO) and tropical cyclones. For the methodological parameters and region chosen, the study finds that WMTEs are most prevalent during in January-February and are associated with a majority of rainfall in Tanzania during this period. Previously published relationships between WMTEs and the MJO are confirmed with this methodology. The results provide a more rigorous analysis than previous work of the connection between WMTEs and tropical cyclones, and finds there to be an association.
This manuscript is one of the best written and most thorough of any paper I've reviewed – thank you to the authors for making the review such an easy process. I consider the application of atmospheric river methodology to provide a useful step forward in rigour, and to have enabled the authors to provide some useful quantification of the phenomenon. I have no major objections to publication, but there's a few aspects that I think the authors need to clarify – there were a few bits of text where it was difficult to understand how the conclusions related to the presented figures. I outline these points below.

Thank you very much for your thorough review of our work and for your positive assessment of our manuscript. Please find below our responses to each of your comments.

 L91 – 45 degrees. Is this plus/minus 45 or plus/minus 22.5 degrees. Please clarify.

This is plus/minus 45 degrees. We have updated line 91 to read "the direction of the moisture transport vector was within the range +/-45 degree of westerly"

 L108 – The lat range used is quite a bit further south than in Finney et al. Whilst the sentence does not say anything incorrect, I think it is worth stating the difference clearly so that readers know to expect that there may be some fundamental methodological reasons for differences in results to Finney et al.

We provided more clarity about the differences in methodology at L108: "Though our study area includes the that of Finney et al. (2020), our results are not necessarily expected to be in agreement, because events in that study were detected using area averaged integrated vapour transport anomalies while here, WMTEs are detected by identifying specific contiguous regions where 700 hPa moisture transport met magnitude and direction criteria. Further, since we use a larger latitude range for our study region, our events did not necessarily affect their study region at all.

Sec2.2 - I'm would have liked supp text S1 to have been referenced in the methods section. I was pleased to stumble across it later in the paper, but it's really at this point that I want to dig into sensitivities.

We moved the reference to S1 to the methods section at line 94.

Sec2.2 - sensitivities to 500km distance to TC have not been discussed? Please share any findings you already have on this, and at least acknowledge this parameter choice when discussing the TC results in 3.2. In particular, the result in Finney et al. of suppressed occurrence of westerlies around Lake Victoria when TCs are in Mozambique channel would not be seen when using this constraint (I think). Please consider discussing this point more fully. Whilst, I think the authors have done a nice job analysing this, I think some further explaoration is needed to pick apart some of the regional details that might matter.

We performed a sensitivity analysis of how the fraction of TC-WMTEs changes as the distance threshold varies from 250 km to 750 km and found that each 250 km change in the distance threshold altered the number of TC-WMTEs going over the EEA line by only 3%. We added a description of this sensitivity analysis in the text at line 101 and included more details in the supplementary information, including Fig. R2.1.

[Figure]

*Figure R2.1. Percentage of WMTE days that cross the EEA line which were TC-WMTEs, varying the distance threshold from 250 km to 750 km*

Sec2.3 - similarly to above comment, some sharing of any findings regarding sensitivity to precip attribution parameters, e.g. maxima separation distance.

We have added a description of findings regarding the sensitivity of these parameters at line 122. We also include description of a repeat analysis that we performed for R1 where we only attributed precipitation inside the WMTE mask. The new text reads as follows:

The sensitivity of this approach is reported in Konstali et al. (2024): the method is almost insensitive to changes in the wet-day threshold, and while it was more sensitive to changing the distance threshold from 500–1000 km, the method is half as sensitive as comparable approaches which attributed precipitation out to a fixed distance threshold away from the weather feature. For comparison, we repeated our analysis using the most conservative possible approach: only attributing precipitation inside the WMTE mask. Using this approach, precipitation amounts inside the region where WMTEs were common were reduced by around 50%, and as expected, precipitation outside this region decreases considerably (Fig. S8). However, we chose to retain the approach of Konstali et al. (2024) as past work has shown that on days with the westerly flow, there is often enhanced precipitation beyond the margin of the main region with westerly moisture transport (e.g., Kilavi et al., 2018; Collier et al., 2019).

Fig1 – It would be nice to see a range of different kinds of examples (including some that seem not to have worked quite as well) in order for readers to get a feel for this method. It's nice to have an example showing it worked nicely, but it's a new method so a few more in a supplementary figure would be appreciated.

We have added some more examples of events to the supplementary information: Fig R2.2 shows several days during a large precipitation event in the anomalously wet long rains of 2018 and Fig. R2.3 shows a more diverse range of examples. The text has been updated at line 124 as follows:

"More example detections are available in the SI, including a large precipitation event that persisted over multiple days during the anomalously wet long rains of 2018 (Fig. S4) and a variety of other examples providing some insight into the method (Fig. S5)."

[Figure]

*Figure R2.2. [New SI Fig.] Example WMTE detections during the long rains 2018. For each day, the left panel shows the 700 hPa moisture transport field along with the masks of detected WMTE masks. The red line is the EEA line that masks must intersect to be described as affecting EEA. The green dot present in (d-f) is TC Eliakim. The middle panel shows the daily ERA5 precipitation total and the precipitation objects. Each colour is a different object. The right plot shows the WMTE masks overlaid with the attributed precipitation.*

[Figure]

*Figure R2.3. [New SI Fig.] Example WMTE detections. For each day, the left panel shows the 700 hPa moisture transport field along with the masks of detected WMTE masks. The red line is the EEA line that masks must intersect to be described as affecting EEA. The green dot present in (d-f) is TC Eliakim. The middle panel shows the daily ERA5 precipitation total and the precipitation objects. Each colour is a different object. The right plot shows the WMTE masks overlaid with the attributed precipitation.*

We also included the following text in the supplementary information:

- Events had considerable variation in latitudinal and longitudinal extent. Some events were extremely wide (e.g. Fig. R2.3a), while others were narrower (e.g. Fig. R2.3b). Events usually extended into the Indian ocean, but sometimes extended further west (e.g. Fig. R2.3b)
- Most of the objects we detected showed very coherent moisture transport. But the very rare detections in JJAS tended not to be such structured objects, often appearing as a patchwork of areas of westerly transport (e.g. Fig. R2.3c, d). However, we inspected the set of detected events and confirmed that such events are extremely rare in the main months of interest in this study.
- The threshold approach occasionally resulted in seemingly coherent events being split into separate entities (e.g., Fig. R2.3e, f). In Fig. R2.3f, TC Galy is active off the coast of Madagascar, and there is a WMTE over EEA. There appears to be westerly moisture transport between the WMTE and the TC-WMTE associated with TC Galy, but the transport in this region did not exceed the 70th percentile threshold. It appears that these objects could be described by a single TC-WMTE. However, since the precipitation did not extend across the two objects, it is possible that the WMTE over EEA has a different driver.
- These examples also raise some questions about precipitation attribution: Fig. R2.2a and f show how on some days, especially when multiple westerlies were detected in the region, or when other convective systems like TCs were active, the region with attributed precipitation could extend very far from the edge of the WMTE mask. We chose to use this attribution method because previous work has shown that even areas rather distant from the main area of westerly transport experience enhanced precipitation on days with enhanced westerly moisture transport, and so this approach allows us to identify moisture that is associated both with WMTEs and interactions between WMTEs and other systems like the MJO.

Fig1 caption. "black dashed line" - I can't see this well. I suggest maybe making it red, or something similar to make it stand out more.

We changed the colour of the line in the figure to red.

L126 / Fig2 caption – I don't think this figure and results are based on the constraint of WMTEs crossing the EEA line? Otherwise panels such as fig2d wouldn't make sense because they ahve no events over that line. Can you it clear that these maps are independent of that constraint.

Thank you for the clarification. We amended the figure caption to read: "Statistics of WMTEs detected in the whole domain: …"

 Fig2 – have you considered including a metric for interannual variability? I imagine some years there might be quite a few more instances, and in some years no instances. Perhaps an IAV metric could be added as another row to indicate where such variability is high? Possible metric that could indicate something useful related to this... 95th percentile of yearly WMTEs events.

We added an extra row to Fig. 2 showing the 95th percentile number of days per month during a season with a WMTE present. In JF, we observed events persisting for up to 20 days.

In response to R1, we will add a "feature tracker" that will enable to look at the individual lifetime of a WMTE, rather than how long there are events present at a certain location.

Note that when we were making the new version of this figure, we found that we also see roughly one WMTE day per month in OND, which was not present in the version of the figure in the preprint, due to a mistake in the plotting code. This does not cause significant changes to our findings, as it means that we see events in MAM and OND at a similar rate, as part of the peak in JF. However, to retain consistency, we decided to add OND to Fig. 3.

[Figure]

*Figure R2.4. [New version of Fig. 2 in text] Statistics of WMTEs detected in the whole domain: (a–d) typical number of WMTE days per month. (e–h) 95th percentile number of WMTE days per month in a single year. (i–l) typical number of WMTEs per month, defined as periods of consecutive days where there was a WMTE present at that location. (m–p) mean duration of WMTEs. (q–t) median duration of WMTEs, in different seasons. White shading indicates areas where no events were detected and so no mean or median could be calculated.*

L143– can you introduce to the reader why you are looking at the curl.

At line 143, we added "To identify possible connections between WMTEs and cyclonic activity, we also show the curl of the moisture transport field."

L152 – TC_WMTE looks to have more intense curl and vectors than noTC_WMTE. Is this true? And is it significant? If so, could you mention it. Possibly showing a 4 column with the difference of these two would help see that more clearly.

We looked into these differences and also into the differences between WMTE and no WMTE days and decided to replace Fig. 3 with Fig. R2.5

Beyond the discussion already in the paper, based on the revised figure, we will add the following information to the results section

- Days with a WMTE have significant westerly moisture transport **anomalies** extending west and north from the region where WMTEs were particularly frequent
- On TC-WMTE days, westerly transport over EEA tended to be stronger and the region with enhanced westerly transport extended eastwards into the Indian Ocean. As expected, TC-WMTE days had stronger curl over the Indian Ocean, especially in the Mozambique channel, but these differences were only significant in JF.

[Figure]

*Figure R2.5. [New version of Fig. 3 in main text] Composites of 700 hPa moisture transport in (a–d) OND, (e–h) JF, and (i–l) MAM. (a, e, i) Days without a WMTE crossing the EEA line (2◦ N–12◦ S along 29◦ E), shown as the green line. (b, f, j) Days with a WMTE crossing the EEA line. (c, g, k) Difference between the composite of days with, and without, a WMTE crossing the EEA line. (d, h, l) Difference between the composite of days with a TC-WMTE crossing the EEA line and days with a noTC-WMTE (a WMTE that was not within 500 km of a TC) crossing the EEA line. The shading represents the curl of the moisture transport field. The numbers labelled in each figure show the average number of days per month in that composite. Differences are shown only where they were significant at the 95% level under a permutation test.*

Fig3 – Whilst I think you have included the right main figure for your paper, there is part of the dynamics that is left out that you might like to add as supplement. Finney et al fig5 shows an intensification of low-level westerlies across the congo on westerly days. Adding a supp fig equivalent to fig3 but with 900 or 850hPa moisture flux or winds, might add a complimentary view of what goes on dynamically during WMTEs.

[Figure]

*Figure R2.6. [] (as per Fig. 3 in main text but for 850 hPa) Composites of 850 hPa moisture transport in (a–d) OND, (e–h) JF, and (i–l) MAM. (a, e, i) Days without a WMTE crossing the EEA line (2◦ N–12◦ S along 29◦ E), shown as the green line. (b, f, j) Days with a WMTE crossing the EEA line. (c, g, k) Difference between the composite of days with, and without, a WMTE crossing the EEA line. (d, h, l) Difference between the composite of days with a TC-WMTE crossing the EEA line and days with a noTC-WMTE (a WMTE that was not within 500 km of a TC) crossing the EEA line. The shading represents the curl of the moisture transport field. The numbers labelled in each figure show the average number of days per month in that composite. Differences are shown only where they were significant at the 95% level under a permutation test.*

Thank you for the suggestion. We made another version of revised Figure 3 for the supplementary information that shows the 850 hPa fields instead, shown in Fig. R2.6. We have added this figure to the SI and will add the following information in the results section:

- low level westerlies over the Congo are also enhanced during WMTE days compared to days without a WMTE
- In OND and MAM, low level anomalies also extend further west over the Atlantic. This is slightly different to the upper level, where anomalies extend further west only for MAM.
- Low level differences between TC-WMTEs and WMTEs are similar to the upper level: TC-WMTEs typically have stronger westerly transport over the Congo compared to noTC-WMTEs.

Fig 3 caption - "noTC-WMTE" is a bit confusing, I initially read it as "no(TC-WMTE)". I don't think you need to change it. But can you spell it out in the cpation instead of just saying a "noTC-WMTE crossing.."

We do define the "noTC-WMTE" acronym in line 98 in the methods section, but for clarity we have also amended the caption to describe noTC-WMTE as "a WMTE that was not within 500 km of a TC"

Fig4 hatching – I feel like it would be more useful to show where there is a significant difference compared to inactive days.

Thank you for the suggestion. We used Fisher's exact tests to see whether the total number of WMTE days at each grid box in that season during the set of MJO phases was significantly different from inactive MJO days. We performed one tailed tests and tested both alternative hypotheses, to highlight areas where, for a particular season, the number of WMTE days in those particular MJO phases was higher or lower compared to inactive days, with the 95% confidence level, including a Benjamini/Hochberg false discovery rate correction. In the new figure, dot (cross) hatching in each panel shows where WMTE days were more (less) common on days in that season on those MJO phases, compared to the inactive phase. It shows that the favourable phases significantly increase the rate at which WMTEs occur over large parts of EEA in OND, JF, and MAM, while decreases in the unfavourable phases are mostly insignificant over EEA except over limited regions. We have changed replaced Fig. 4 in the text with Fig. R2.7 and will add this information to the results section.

[Figure]

Figure R2.7. The relationship between MJO phase and the occurrence of WMTE days. The number of days per month with a WMTE in each grid box during: (a, e, i) JF; (b, f, j) MAM; (c, g, k) JJAS; (d, h, l) OND. (a–d) show favourable MJO phases (phases 2–4); (e–h) show unfavourable MJO phases (phases 1 and 5–8), and (i–l) show inactive MJO. The number on each panel indicates the average number of days per month in the MJO phases shown in that panel. Cross (dot) hatching shows areas where the rate of WMTE occurrence was significantly lower (higher) than the rate under inactive MJO using single tailed Fisher exact tests at the 5% level. The red dashed line shows the 'EEA line.'

L210 - "throughout the year" - I've struggled to follow how you can say this when your fig5 shows the full year and JF, but the full year is dominated by JF, I think? To say throughout the year, you'd need to look at MAM, JJAS and OND separately in your supplementary figures. This would be interesting to see, and would probably help compare to previous

results, especially if you looked at MAM. I think you need adjust the text, if you don't analyse MAM on it's own.

We agree with your comment, we have amended line 210 to be "at least in JF"

We also repeated the analysis for MAM (Fig. R2.8) and OND (Fig. R2.9) and include these in the supplementary (Fig. S7-8). We have amended the main text at line 203 to be:

"In both MAM (Fig. S7) and OND (Fig. S8), the signal was much noisier, but in both seasons there were large regions in the central Indian Ocean, particularly during favourable MJO phases, where the risk ratio exceeds one. In MAM, the same is true for the inactive phases. Even for JF, the values are noisy with strong local fluctuations of opposing sign, making it hard to draw strong quantitative conclusions, but the analysis suggests that from October to May, TC presence is associated with an enhanced rate of WMTEs in most of the south-west Indian Ocean and during favourable MJO phases. In JF and MAM, the enhanced rate is also seen during unfavourable MJO phases and, in JF, when the MJO is inactive."

[Figure]

Figure R2.8. As Fig. 5 in main text, but all panels show MAM. How the presence of a tropical cyclone at different locations changes the probability of a WMTE day in EEA. (a–c) The probability of a WMTE crossing the line at 29°E from 2°N to 12°S, shown by the black dashed line, given the presence of a TC in 5° grid boxes. (d–f) Relative risk for each box, showing the ratio of the probability of a WMTE crossing the line given the presence of a TC, compared to the probability of a WMTE crossing the line, given there is no TC anywhere in the Indian Ocean. Boxes with a dot have at least 5 TC reports in the period 1980–2022. (a, d) MJO phases 2–4, (b, e) MJO phases 5–1, (c, f) MJO inactive

[Figure]

*Figure R2.9. As Fig. 5 in main text, but all panels show OND. How the presence of a tropical cyclone at different locations changes the probability of a WMTE day in EEA. (a–c) The probability of a WMTE crossing the line at 29°E from 2°N to 12°S, shown by the black dashed line, given the presence of a TC in 5° grid boxes. (d–f ) Relative risk for each box, showing the ratio of the probability of a WMTE crossing the line given the presence of a TC, compared to the probability of a WMTE crossing the line, given there is no TC anywhere in the Indian Ocean. Boxes with a dot have at least 5 TC reports in the period 1980–2022. (a, d) MJO phases 2–4, (b, e) MJO phases 5–1, (c, f ) MJO inactive*

L215-224 - There's also a slight other issue with your comparison to previous studies, because you have defined WMTE quite far south compared to the focus regions of previous studies. You may see a positive relation to WMTE because there is a WMTE over tanzania, but that WMTE is not spreading over Lake Victoria. I think you need to acknowledge this, and at least highlight the need for futrher research to pick it apart.

Good point. We have changed the text at line 221 to read: "Differences in the long-term patterns of WMTE, MJO and TC interactions likely arise because most of the WMTEs we detected were relatively far south compared to the study regions of previous studies. This highlights the need for further work exploring spatial variations in the relationship between precipitation, westerlies, and TCs in the EEA region."

 L246 – Finney et al showed that westerly day rainfall was upto 200% of daily average rainfall. I would call this up to 100% increase, as daily rainfall would be 100%. Check your sure of your phrasing.

Thank you for pointing this out. We amended the text to read 100% increase.

Technical comments

 L261 – typo – capital "S" on "WMTES"

Thank you for spotting this. We have amended the text

---

## Author Comment (AC2)

**Responses to Reviewer 1**

Reviewer comments are provided in black font, while our replies are in blue.

This article analyzes events of westerly moisture transport across Africa (and nearby regions), and links those events to the phase of the Madden-Julian Oscillation (MJO), and the presence (or lack) of tropical cyclones (TCs) over the Indian Ocean. It finally provides an estimation of the precipitation that can be attributed to those westerly moisture transport events (WMTEs).

The authors developed a detection algorithm, quite similar to those used to monitor atmospheric rivers, to identify the WMTEs. The algorithm is unique in that ARs have a strong poleward component, while inter-tropical WMTEs are mostly zonal. The approach is sound and there was a clear need to have more in-depth understanding of those WMTEs.

The following analyses are less convincing. They remain very descriptive. The authors build samples consisting of different phases of the MJO, combined with the presence or absence of TCs, but the main object proposed in this work (the WMTEs) are not analyzed with more detail than just their presence or absence. Another possible issue comes from the attribution of rainfall to WMTEs: the justifications are not very clear but I understand (Fig. 1) that precipitation extending far beyond the contours of the detected WMTEs can be considered as linked to it. This is clearly not what is usually done when working with those detected moisture transports.

For those reasons, I believe that there is a very good study to be done with those WMTEs, but I think that the current version of the manuscript required major modifications and improvements before it can be accepted for publication. Even though I recommend major corrections, I would like to encourage the authors — there are really great ideas here, we'd just need more physical characterizations (and understanding) of the WMTEs, their properties, and their mechanisms (see detailed comments below).

Thank you very much for your thorough review of our work, for your positive assessment of our manuscript, and for your constructive feedback for improvement. Please find below our responses to each of your comments.

Major comments

1. What are the WMTEs, physically speaking? ARs have been described as "A long, narrow, and transient corridor of strong horizontal water vapor transport that is typically associated with a low-level jet stream ahead of the cold front of an extratropical cyclone" according to the AMS Glossary. This is because ARs are a concept mostly used for extra-tropical climate and weather, where transient disturbances (atmospheric highs and lows) have fronts that separate air masses, where baroclinic instability develops. This definition (and even the schematics that are often provided in some studies) are useful to conceptualize the AR objects, from a physical point of view. The concept of WMTEs strongly differs, even if it is detected by algorithm that are not so different. This is not only because WMTEs are more zonal, but also, because they develop in a tropical climate, where vorticity is very low because of the proximity to the equator. In those regions, large-scale cellular

circulations (that are, partly, conceptual objects) can develop (including zonal ones), as part of the MJO at the intraseasonal timescale, or ENSO at the interannual timescale. Are WMTEs linked, or even part, of those large-scale cells? Do they correspond to "bursts" or transient increases in their lower branch, when moisture convergence is located further east over the Indian Ocean sector (since you use a quite high threshold to define them, you only depict the most intense phases of those westerly circulations)? Do the WMTEs help refine (regionally) the conceptual schemes of the MJO as proposed by Madden and Julian (e.g., Fig. 3 of their 1994 paper, cited in this work), or propose more detailed vertical cross-sections? So, overall: what are WMTEs, physically speaking?

The same could be said about the interannual timescale, especially in OND when the EEA region experiences the Short Rains whose interannual variability is strongly tied to the Walker circulation (= are WMTEs partly driven by changes in the Walker-type circulation ?). See e.g. an (already old) paper by Hastenrath about the detection of those cellular circulations: Hastenrath, S (2000) Zonal circulations over the equatorial Indian Ocean : Journal of Climate 13, 2746-2756.

When reading the article I had a similar question, about their interpretation, when the dominant winds are easterly (usually the case where/when trade winds are well developed) vs. the regions where westerlies prevail (like the northwestern Indian Ocean in boreal summer, due to the monsoonal circulation). In other words, how can the threshold used differentiate transient from seasonal circulations. But the questions raised above are more general and encompass that particular case. All these questions are, in my opinion, super important, to know what are the objects we're talking about, and that we consider so extensively in this work.

Thank you for these insightful comments about the nature of WMTEs. We agree that the paper would benefit from additional discussion about the physical nature of these events, which we will add to the revised manuscript. We note that we developed the detection framework as part of a project focussed on understanding extreme snowfall events on Kilimanjaro's glaciers, and to investigate the relation between extremes, WMTEs, the MJO, and TCs. Therefore, providing a definitive answer about the causes of WMTEs is outside the scope of our study but will hopefully be facilitated in future work by our objective timeseries and preliminary analysis

The discussion we will add to the manuscript addressing the main aspects of your comments includes:

1) **Connection to large scale cellular circulations**: it has already been proposed that EEA westerlies are part of the westerly transport into the main area of MJO convergence (e.g., Pohl & Camberlin (2006)). Furthermore, the seasonal cycle of our events is consistent with the seasonal cycle of the MJO: MJO tends to be stronger, with more organised events in DJFM, with a peak in JF (Zhang & Dong, (2004)). Therefore, some WMTEs may potentially be part of the western branch of cellular circulation patterns associated with the MJO.

2) **Connection to the Walker circulation**: the Indian Ocean Walker-like circulation tends to be associated with westerly winds over the central equatorial Indian Ocean rather than directly over EEA, and its influence on EEA rainfall has been mainly been

emphasised for the OND season (Hastenrath, Polzin, & Camberlin (2004), Hastenrath & Polzin (2005), Hastenrath, Polzin, & Mutai (2010)), while our analysis identifies WMTE activity mainly in JF, which we think is related mainly to the MJO activity peak.

3) **Differentiating transient from seasonal circulations**: we showed that the median length of time for a WMTE to persist at a single grid cell is roughly 1-3 days in EEA, and for reviewer 2 we have also added panels showing that in most of EEA, the 95th percentile number of days in a single month with a WMTE peaked at 16-20 days in JF, and therefore believe we identify transient rather than seasonal features. In response to your comment regarding metrics characterising WMTEs, we will add feature tracking in the revised manuscript, which will allow us to accurately calculate the lifetime of these events and more robustly assess their persistence. We also note that the 70th percentile threshold, which varies each month of the year and at each grid box, is somewhat lower than the thresholds normally used in AR studies but should still be high enough to exclude purely seasonal features.

In addition to your suggestions, it has also been proposed that westerlies in EEA are simply associated with a reversal of the zonal pressure gradient across the African continent (also in Pohl and Camberlin (2006)) which could be related to several physical drivers, e.g. the presence of multiple TCs, MJO activity, or persistent continental lows (Webster (2019))

We will add discussion about all of these aspects to the revised manuscript.

2. There are no metrics here to characterize the WMTEs, like their length, width, tilt / direction, duration, integrated moisture transport (and location of the maximum), … The AR community produced tens of articles showing that those descriptors are important to better analyze their impacts on rainfall, and help better understand the mechanisms responsible for rainfall, and its space-time variability (including daily amounts or even extremes). In addition, relating rainfall to the location of the outflow boundary of the WMTEs might give potentially interesting results; similarly, the inflow location might give insight into the moisture sources.

This is a nice suggestion and we agree some analysis of these metrics would give some interesting results. We will therefore implement a feature tracker that will allow us to connect overlapping WMTEs from consecutive days and to analyse some of their properties and relations to the precipitation response.

3. Attribution of precipitation to WMTEs. I understand you've considered the Boolean union of the WMTE and Precipitation > 1mm.day-1 contours and attributed all rainfall falling within that new contour to WMTEs. This would mean that precipitation occurring outside the WMTE contour is yet attributed to it. This would be clearly an issue, especially since WMTE can be linked to the MJO that promoted the development of large-scale convective clusters (some of which can reach 10,000km diameters). This would imply you could attribute MJO-caused precipitation to WMTEs. Considering the Boolean intersection instead of the union would certainly decrease the contribution of WMTE to rainfall totals, but the approach would be more conservative and more robust. This is the one traditionally used by the AR community. Yet, double (or triple!) counts are still possible (i.e., in the worst-case scenario, attributing rainfall to WMTEs, MJO and TCs). See e.g. Dacre's papers about the interactions between

atmospheric rivers, warm conveyor belts and cyclones: 10.1038/s41612-025-00942-z; 10.1029/2023jd040557; 10.1175/JHM-D-18-0175.1.

Although the precipitation attribution approach we used was developed for precipitation attribution to various drivers including atmospheric rivers (Konstali et. al., 2024), we repeated our analysis using a conservative attribution approach that only attributes precipitation inside the WMTE mask. Using this approach, precipitation amounts inside the region where WMTEs were common are reduced by around 50%, and as expected, precipitation outside this region decreases considerably (Fig. R1.1). However, we believe that the approach of (Konstali et. al., 2024) is reasonable and provide the following discussion in the text at line 122:

"For comparison, we repeated our analysis using the most conservative possible approach: only attributing precipitation inside the WMTE mask. Using this approach, precipitation amounts inside the region where WMTEs were common were reduced by around 50%, and as expected, precipitation outside this region decreases considerably (Fig. S8). We note that the approach of Konstali et al. (2024) may occasionally lead to over-estimates of attribution (see for example Fig. S5a, f), particularly when other features such as the MJO or TCs are also present in the region. However, we chose to retain our original approach as past work has shown that westerly moisture transport is often associated with enhanced precipitation beyond the margin of the main region with enhanced moisture transport (e.g., Kilavi et al., 2018; Collier et al., 2019) and therefore believe that this provides a reasonable estimate of the impact of WMTEs."

[Figure]

*Figure R1.1. Comparison of precipitation attribution approaches. (a-d) The ERA5 average monthly precipitation from 1998 to 2022 in each season. (e–h) The fraction of precipitation that was attributed to WMTEs in each season using the method of (Konstali et al., 2024). (i–l) The fraction of precipitation that was attributed to WMTEs that lay inside the WMTE mask. (m–p) Difference between (i–l) and (e–h). (q–t) Fractional change between (m–p) and (i–l). Small increases in the amount of attributed precipitation in very dry areas are due to removing the wet day threshold for the conservative approach.*

4. Relationship between WMTEs and TCs: is it a "chicken and egg" problem? Do the WMTEs feed TCs with moisture, or do WMTEs respond to the development of TCs (or convection, more generally) through ageostrophic circulations? While the causality itself

might deserve dedicated studies, it would be quite straightforward to see whether the WMTEs develop before, or after the TC. This could be useful as a first clue to understand how both circulations behave and interact.

This is a very nice question, which we will answer and discuss in the revised manuscript once we implement the feature tracking.

Minor points. There are not many of them because I mostly focused this first review on the main points listed above.

l. 92. How precisely is the 70th percentile of moisture transport calculated? By considering both signs (i.e. easterly and westerly), or just westerly transport occurrences?

The 70th percentile is computed considering both signs. We amended line 92 to make this clear: "the magnitude of moisture transport exceeded the 70th percentile of all magnitudes recorded at that location, for that month, from 1980 to 2022 (including non-westerly days)"

Figure 2. If the main interest in this work is to assess westerly moisture transport across Africa and reaching the EEA region, then the choice of the domain is a bit strange — shifted eastwards, and giving more importance to the Indian Ocean region. Previous work (e.g. on the regional influence of the MJO) discussed zonal moisture convergence between the Congo basin and the Indian sector, so the WMTEs of interest, advecting moisture towards EEA, should be placed over inter-tropical or equatorial Africa. Moisture sources might be continental (Congo basin) or oceanic (Atlantic sector). Why such an eastward-shifted domain? WMTEs have lesser importance for EEA if they occur east of it?

The domain used for WMTE detection (22° W to 102° E and from 40° N to 37° S) includes both the Congo basin and a part of the Atlantic sector. We also extended the domain eastwards so that we could investigate the interaction of WMTEs with TCs anywhere in the Indian Ocean. Most of the figures in the paper do not include the Atlantic part of the domain because the identified WMTEs did not occur there. We briefly mentioned this in the preprint at line 83: "The area used in our detection algorithm included all of Africa and the Indian Ocean, extending from 22° W to 102° E and from 40° N to 37° S. However, we focus our results and discussion on the regions of eastern Africa and south-west Indian 85 Ocean where our phenomenon of interest occurs", but we will add a sentence in the revised manuscript justifying this more clearly.

Figure 3. Why show the curl of the moisture fluxes, rather than their convergence? I'm not saying this is wrong, but this needs to be explained. Moisture convergence may make more sense for rainfall analysis. Vorticity is certainly more meaningful when assessing the links with cyclogenesis.

We show the curl to demonstrate the connection of westerly events to flow curvature over East Africa and cyclonic activity in the SWIO. We have amended line 142 to make this clearer: "To identify possible connections between WMTEs and cyclonic activity, we also show the curl of the moisture transport field."

The quantity we show is therefore equivalent to the relative vorticity but for the moisture transport. We will check the relative vorticity field in case it makes any significant change to the interpretation.

l. 156. TCs themselves are not independent of the MJO. The authors did not discuss this point.

Bessafi, M. & Wheeler, M. C. (2006) Modulation of south Indian Ocean tropical cyclones by the Madden-Julian Oscillation and convectively coupled equatorial waves. Mon. Weather Rev. 134, 638–656

Klotzbach, P. J. (2014) The Madden-Julian Oscillation's Impacts on Worldwide Tropical Cyclone Activity. J. Clim. 27, 2317–2330

Diamond, H. J. & Renwick, J. A. (2015) The climatological relationship between tropical cyclones in the southwest pacific and the Madden-Julian Oscillation. Int. J. Climatol. 35, 676–686

Thank you for pointing out this association which was part of our motivation of breaking down the TC analysis by MJO phase. We have added the following text at line 172: "TC activity is itself influenced by the MJO phase: Liebmann et al. (1994) showed that TC genesis in the Indian Ocean is enhanced when the centre of MJO convection lies over the Indian Ocean (phases 2–4), while Klotzbach (2014) showed that cyclogenesis in also enhanced, particularly in the eastern Indian Ocean, during phase 5. Since we expected TCs further east to have less impact on WMTEs over EEA, we controlled for the effect of MJO on TC genesis by subdividing our analysis using the same classification as in previous sections, with phases 2–4 described as favourable and 5–1 as unfavourable."

Figure 6. What is the use of defining an EEA region like in Finney et al. if it's not used in this work, e.g. to compute regional indices?

We included the EEA region from Finney et al. because it is the most comprehensive existing study of this interplay between MJO, TCs, and westerly moisture transport in driving EEA precipitation and we wanted to compare our findings. We have clarified that we show the box only to allow comparison between the studies in the text and added some discussion of the differences in methodology and results between the studies

**References**

Hastenrath, S., & Polzin, D. (2005). Mechanisms of climate anomalies in the equatorial Indian Ocean. *J. Geophys. Res., 110*, D08113.

Hastenrath, S., Polzin, D., & Camberlin, P. (2004). Exploring the predictability of the "short rains" at the coast of East Africa. *Int. J. Climatol., 24*, 1333--1343.

Hastenrath, S., Polzin, D., & Mutai, C. (2010). Diagnosing the droughts and floods in equatorial East Africa during boreal autumn 2005-08. *J. Climate, 23*, 813--817.

Webster, E. M. (2019). *A synoptic climatology of Continental Tropical Low pressure systems over southern Africa and their contribution to rainfall over South Africa.* Master Thesis, University of Pretoria, Pretoria.

Zhang, C., & Dong, M. (2004). Seasonality in the Madden–Julian Oscillation. *J. Climate*(17), 3169–3180. doi:https://doi.org/10.1175/1520-0442(2004)017<3169:SITMO>2.0.CO;2

---

## Author Response (AR2)

**Response to second review comments from Benjamin Pohl**

Thank you for your positive feedback about our revisions and for your suggested discussion points. We have amended the text at line 369 to incorporate some of your ideas for discussion:

Areas for future work highlighted by this research include: (i) investigating the causality chain and exact processes linking WMTEs, TCs and MJO activity and their variability across the 370 EEA region; (ii) assessing which TC properties, such as location and intensity, may affect their association with WMTEs; (iii) more spatially detailed analysis of how WMTE properties modulate their association with extreme precipitation in EEA; (iv) analysis of how WMTEs and their impacts on rainfall are modified by phenomena previously linked to EEA precipitation variability, such as the El Niño Southern Oscillation and the Indian Ocean Dipole (e.g., Palmer et al., 2023) and Atlantic sea surface temperatures (e.g., Ward et al., 2023); and, (v) further sensitivity analysis of how the choice of the parameters used to 375 identify WMTEs affects the attributed EEA precipitation.